# Ribosome Structural Changes Dynamically Affect Ribosome Function

**DOI:** 10.3390/ijms252011186

**Published:** 2024-10-17

**Authors:** Lasse Lindahl

**Affiliations:** Department of Biological Sciences, University of Maryland, Baltimore County (UMBC), 1000 Hilltop Circle, Baltimore, MD 21250, USA; lindahl@umbc.edu

**Keywords:** translation, ribosome types, specialized ribosomes

## Abstract

Ribosomes were known to be multicomponent complexes as early as the 1960s. Nonetheless, the prevailing view for decades considered active ribosomes to be a monolithic population, in which all ribosomes are identical in composition and function. This implied that ribosomes themselves did not actively contribute to the regulation of protein synthesis. In this perspective, I review evidence for a different model, based on results showing that ribosomes can harbor different types of ribosomal RNA (rRNA) and ribosomal proteins (r-proteins) and, furthermore, need not contain a complete set of r-proteins. I also summarize recent results favoring the notion that such distinct types of ribosomes have different affinities for specific messenger RNAs and may execute the translation process differently. Thus, ribosomes should be considered active contributors to the regulation of protein synthesis.

## 1. Introduction

Proteins are the most abundant type of macromolecule in biological cells. They contribute to all cell functions, from structure to metabolism, cell growth, differentiation, and communication. Proper cell function requires this multitude of proteins to be synthesized in physiologically appropriate amounts. Many investigators have been fascinated with the idea that different classes of ribosomes with different make-ups and functional characteristics are essential for obtaining this balance. This view was supported by ultracentrifugation experiments revealing ribosome variants that lack specific ribosomal proteins (r-proteins) [1] and electron microscopic observations showing diverse ribosome morphologies [2]. However, the viewpoint of ribosome homogeneity prevailed, since ribosome heterogeneity could also be attributed to imperfect techniques for preparing ribosomes. Thus, the prevailing view was that all ribosomes would translate any messenger RNA (mRNA) with a proper translation initiation site.

Nonetheless, ribosome structural heterogeneity has remained a topic of discussion. Over recent decades, new evidence has shown that ribosomes of different compositions are translationally competent, but selectively translate different mRNAs (often called “specialized ribosomes”). However, this concept has remained a topic of debate [3]. In this Perspective, I discuss recent experiments examining how rRNA and r-protein composition affect ribosome function. This leads me to assert that variations in ribosome composition and structure contribute to the regulation of translation.

## 2. Basic Organization of Ribosomes

Ribosomes throughout all biological domains consist of two unequal subunits built with ribosomal RNA (rRNA) and r-proteins (Table 1). rRNA is generated from long primary transcripts by RNase processing simultaneously with r-protein incorporation into precursor ribosomes and rRNA modification under the supervision of ribosome assembly factors. In eukaryotes, the assembly process begins in the nucleolus, continues in the nucleoplasm, and is completed by exporting the precursor particles to the cytoplasm, where final maturation occurs. The assembly process is different in bacteria and eukaryotes. Still, it is largely conserved in eukaryotes, even though new assembly factors were added during evolution [4,5].

During ribosome construction, the rRNA is folded into universally conserved secondary structural elements that organize the rRNA into self-containing domains (Table 1) [6,7]. The rRNA structure was expanded throughout the evolution of eukaryotes by the insertion of RNA elements, called expansion sequences (ESs), that “elaborate” on the secondary structures while maintaining the universal domain organization [6,8,9,10,11]. This process explains why human rRNA is about twice as long as bacterial rRNA (Table 1). Most ESs are placed at the surface of the ribosome and are typically found on the ribosome “solvent side”, facing the cytoplasm in the translating 80S ribosome, and are thus available for interaction with other cellular components [9]. A smaller number of ESs contribute to forming new and altered “bridges” between the subunit “interfaces” that face each other during translation [12]. Ribosomal proteins evolved both before and after the major biological domains separated, resulting in some r-proteins being present in all organisms (universal), while others are found only in some evolutionary lineages (bacterial, archaeal, eukaryotic) (Table 1) [9,13,14].

**Table 1 ijms-25-11186-t001:** Organization of cytoplasmic ribosomes.

	Small Subunit						Large Subunit					
	rRNA				Proteins [13]		rRNA				Proteins [13]	
Biological domain	Subunit name	rRNA Name	rRNA length (nts)	# RNA domains	# proteins	Protein types	Subunit name	rRNA Name	rRNA length (nts)	# RNA domains	# proteins	Protein types
Bacteria	30S	16S	~1550	3	22	B: 7	50S	23S	~2900	6	33	B: 9
						BAE: 15		5S	~120	3		BAE: 24
Eukaryotes	40S	18S	~2000	3	33	BAE: 15	60S	25-28S	~3380–~4700	6	46	BAE: 18
						AE: 12		5.8S	~150	1		AE: 21
						E: 6		5S	~120	3		E: 7

Protein types B: bacteria; A:Archaea; E: eukaryotes.

Despite the evolutionary divergence, all types of ribosomes are structurally alike and perform the same essential functions. The one significant functional difference is in the mechanism for translation initiation in bacteria and archaea versus eukaryotes [14]. The small ribosomal subunit manages translation initiation and decoding of the mRNA sequence (Decoding Center), while the large subunit is the site where peptide bond formation (Peptidyl Transferase Center) and initial folding of the nascent peptides (Exit Tunnel) occur. Both subunits also interact with many auxiliary factors that assist the ribosome during specific portions of the translation process [14].

## 3. Ribosomal RNA Sequences Affect the Translation Function

### 3.1. Expansion Sequences in rRNA Are Important for Ribosome mRNA Selection and Translation Fidelity

Selection pressures for adaptation to environmental conditions have resulted in the evolution of rRNA paralogous genes. In bacteria and *Archaea*, some species have single-copy rRNA genes, but others harbor several rDNA variants mapping to disparate genomic positions. Furthermore, the genes encoding rRNA for the two ribosomal subunits can be organized as single transcription units, or as unlinked rRNA genes. In contrast, eukaryotic rRNA genes are arranged in one or more tandem repeats, often residing on separate chromosomes [15,16,17]. Comprehensive studies revealed pervasive variations in the genomic rDNA copy number, nucleotide substitutions, and indels in humans, mice, and yeast that can be ascribed to the evolutionary progression necessitated by the evolving demands on ribosome function [18,19,20]. The expression of individual rRNA types varies significantly between individuals, cell types, and tissues. Moreover, it responds to environmental cues and diseases [21,22,23,24]. For example, zebrafish maternal rRNA types are replaced with somatic equivalents during development [25], and the relative expression of rRNA operons in bacteria changes during adaptation to changing growth media, temperature, and stress [20,26].

The functions of the individual rRNA parts are now being dissected. Most eukaryotic rRNA genes cannot be analyzed by standard genetic manipulation because of the large number of rRNA alleles in multiple repeat arrays around the genome. However, *Saccharomyces cerevisiae* (yeast) is a valuable model system for the dissection of rRNA functions. First, rRNA can be expressed from a single plasmid-borne rRNA transcription unit; second, rRNA expression from the endogenous chromosomal genes can be effectively eliminated using a temperature-sensitive RNA polymerase I mutant or deleting the chromosomal rRNA genes, which are located in a single array [27,28,29].

Deletion analysis shows that many ESs are necessary for normal posttranscriptional rRNA processing and ribosome assembly [8,30,31,32]. Interestingly, the products of the incomplete rRNA processing observed in the ES deletion mutants are very similar to the pattern seen after blocking the production of r-proteins surrounding each rRNA ES, suggesting that ES RNA-protein structure elements participate in the ribosome assembly [30].

The yeast rRNA plasmid system is also used to interrogate several rRNA ESs for their contributions to translation initiation, translation accuracy, and mRNA selection. The ESs were extended and modulated during the evolution of eukaryotes so that equivalent structures in different organisms vary in both sequence composition and length. This was exploited to generate yeast strains with a hybrid ES9S, in which part of the yeast ES9S was replaced with the corresponding portion of the human ES9S. An r-protein gene in the hybrid ES9S mutant was then FLAG-tagged, allowing for the purification of ribosomes harboring the ES9S hybrid. Messenger RNA binding experiments showed that the ribosomes carrying the hybrid ES9S preferentially bind to a portion of the 5′ untranslated region (5′ UTR) of mRNA transcribed from the human Homeobox gene 9 (Hox9), while the unmodified yeast ribosomes do not, indicating that the ES9 participates in the selection of mRNA [33,34]. In agreement with this model, structural analysis of the *Trypanosoma cruzi* 43S preinitiation complex reveals that the translation initiation factor eIF3 contacts three different ESs in the 40S subunit (ES6S, 7S, and 9 ESs) [35]. Figure 1 illustrates how some ES hairpins are important for the selection of translation initiation sites.

Expansion sequences are also important for translation accuracy. The rate of miscoding has decreased through evolution, presumably due to the selection of more accurate ribosomes that can faithfully translate the longer reading frames appearing during evolution [36]. Accurate recognition of aminoacyl tRNA and translocation require the movement of universally conserved nucleotides in the codon recognition center. Due to its position in the ribosome, the 18S rRNA expansion segment ES7S was targeted for evaluation of its involvement in aminoacyl tRNA recognition. Deletion or shortening ES7S increases mistranslation, but has little effect on 40S assembly and structure. Moreover, replacing the yeast ES7S with its human equivalent increases translation accuracy, demonstrating a function of this expansion sequence in the translation accuracy [37,38].

As mentioned, the center for accurate codon decoding resides in the small ribosomal subunit. It was, therefore, unexpected that the 60S ES27L expansion sequence also contributes to translation fidelity. This is indicated by the finding that yeast ribosomes carrying a hybrid yeast/human ES27L have higher translation accuracy than wildtype yeast ribosomes. Moreover, deleting the end of the human ES27L in the yeast/human hybrid ribosomes reduces decoding fidelity [39]. It is not clear how ES27L affects translation accuracy, but it is hypothesized to be related to the fact that methionine amino peptidases bind to ES27L close to the end of the peptide exit channel and remove the initiator methionine of some nascent peptides. This may, in unknown ways, interact with the decoding center on the small subunit [39,40,41,42]. Increased decoding accuracy may also result from binding additional enzymes for nascent peptide modification to ES27L and nearby ES7L [40,41,43]. Further complexity in the contributions of ESs is suggested by allosteric changes in the 23S rRNA resulting from physical interactions between different ESs in kinetoplast ribosomes [44,45].

### 3.2. Posttranscriptional rRNA Modification Responds to Growth Conditions and Development and Adjusts Ribosome Function

Both base and ribose moieties of rRNA nucleotides undergo posttranscriptional modification. The most frequent base modification is dimethylation of an adenine close to the 3′ end of the rRNA in the small ribosomal subunit [46]. However, the methylation enzyme (Dim1) also generates single-A6 methylation at lower rates. During sulfur starvation of yeast, the balance shifts toward monomethylation, which makes the ribosomes more efficient at translating mRNAs encoding proteins in sulfur metabolism [47].

Numerous other nucleotides in eukaryotic rRNA are modified by methylation of the 2′ position of the ribosome moiety (2′O methylation) or by conversion of uridine to pseudouridine [48,49]. The methylation is catalyzed by fibrillarin, and the pseudouridylation is catalyzed by dyskerin, each of which forms complexes with two other proteins specific to each enzyme. The targets of these enzyme multiplexes are determined by small nucleolar RNAs (snoRNAs) containing short nucleotide sequences complementary to the target sites [50].

Extensive analysis of 2′O ribose methylation in yeast, humans, and other organisms shows that, while modifications at some sites are (nearly) complete, other sites are only partially and variably modified [51,52,53,54]. The modification pattern changes in response to the expression of specific genes, for example, the MYC proto-oncogene, and exposure to Transforming Growth Factor beta (TGF) [51,55]. Moreover, the methylation patterns can change the ribosome preference for translating specific groups of mRNAs [51,54,56,57,58] and the translation rate of specific codons and short repeat sequences [59].

A recent landmark discovery showed that rRNA 2′O ribose methylation differs between the germ layers in mice [60], suggesting that 2′O ribose methylation is a fundamental element in tuning the ribosome to support the progression of development. Accordingly, the degree of 2′O ribose methylation of specific 28S rRNA nucleotides is critical for human embryonic (hESC) stem cell differentiation [61]. Moreover, ablation of the methylation at 28S:U3904 by deleting the gene for the corresponding snoRNA inhibits the translation of proteins in the Wingless-related Integration Signaling Pathway (Wnt), which is critical for normal development [60]. The 2′O methylation pattern also contributes to coping with stress as it changes during hypoxia, which, in turn, affects the ribosome preference for translation initiation at specific IRES structures [62]. Other recent experiments show that mechanical muscle overload changes the expression of snoRNAs, altering the rRNA 2′O methylation pattern [63]. Given that rRNA methylation occurs in the nucleus, this implies that mechanical stress induces the synthesis of new ribosomes with a different methylation pattern. Studies of *Trypanosome brucei* and *Leishmania* have also implicated single changes in nucleotide pseudouridylation in the accuracy of tRNA recognition and translation of specific mRNAs [64,65]

## 4. The Ribosomal Content of r-Protein Affects the Ribosome Function

### 4.1. The Ribosomal r-Protein Content Is Dynamic and Critical for Ribosome Specificity

It has generally been assumed that ribosome biogenesis and function require a complete set of r-proteins. Indeed, blocking the synthesis of many r-proteins abolishes ribosome assembly and stops the accumulation of new ribosomes [66,67,68]. However, deletion experiments have shown that 22 *Escherichia coli* and 14 yeast ribosomal proteins are not essential, demonstrating that ribosomes can work without a complete set of r-proteins [69,70]. Moreover, many r-proteins exist in two or more paralogous forms, whose expression varies with growth conditions (discussed below).

Several lines of evidence show that the r-protein composition of mature ribosomes is dynamic. First, isotope labeling of proteins demonstrated the exchange of ribosomal proteins in mature cytoplasmic ribosomes in cells from bacteria to neurons [71,72]. Second, r-proteins in mature bacterial ribosomes turn over at different rates [73]. Third, ribosomes damaged by r-protein modification can be reactivated by replacing the damaged r-proteins with newly synthesized proteins [74,75].

The ribosome composition of an r-protein impacts ribosome function in all stages of the translation process, but, so far, most work has been focused on the translation initiation process in both bacteria and eukaryotes. In bacteria, the translation of most genes begins with base pairing between the Shine–Dalgarno sequence (SD) in the mRNA “leader sequence” upstream of the start codon and the anti-Shine–Dalgarno (aSD) in the distal region of the 16S rRNA. r-Protein bS1 facilitates this by unfolding any internal base pairing involving the SD in the free mRNA, thus making the SD available for pairing with the aSD [76,77]. Some rare mRNAs have no leader sequence, so the AUG start codon consists of the first three nucleotides at the 5′ end of the mRNA. Analysis of translation initiation on one such mRNA, encoding the cI protein of phage lambda, shows that ribosomes containing r-proteins uS2/Rps2 fail to bind correctly to the cI initiation codon. In contrast, ribosomes missing this protein initiate translation correctly at the initiation codon. These ribosomes also lack bS21/Rps21, and the absence of the two proteins is thought to ease the propagation of the cI mRNA through the ribosome during initiation [78].

The effects of varying r-protein content in the ribosome have also been studied in eukaryotes. The yeast r-protein e26/Rps26 is located in the mRNA channel and is, thus, of interest for understanding the ribosome-mediated regulation of translation. This was studied by tagging the protein and controlling its expression with a galactose promoter, allowing the separation of ribosomes with and without eS26/Rps26. Comparing the two classes of ribosomes shows that eS26/Rps26 is important for ribosome discrimination between mRNA Kozak elements that differ by only one nucleotide [79]. Furthermore, the eS26/Rps26 protein can dynamically be removed and returned to the ribosome with the help of the dedicated Tsr2 chaperone [80], which is important for both ribosome repair after oxidative damage to eS26/Rps26 [69] and removal of eS26/Rps26 during adaption to salt, sorbitol, and pH stress [81] (Figure 2).

The effect of the r-protein content was also analyzed in human embryonic stem cells by mass spectroscopy quantification using heavy isotope standard peptides as internal standards [82]. While many r-proteins are found in stoichiometric amounts in polysomes, uL1/Rpl10A, eL38/Rpl38, eS7/Rps7, and eS25/Rps25 are underrepresented, indicating that a significant fraction of translating ribosomes lacks these proteins. These proteins are all on the surface of the ribosome and lack extensions penetrating to the ribosome center, suggesting that they are easily removed from the ribosome, but they do not all map to a particular part of the ribosome [83]. Further examination of ribosome-mRNA complexes purified by r-protein tagging shows that separate mRNA pools are associated with ribosomes lacking either uL1/Rpl10A or e25/Rps25 [82]. Moreover, r-proteins eS25/Rps25, RACK1, and eL38/Rpl38 are necessary for IRES-mediated translation initiation on certain mRNAs [84,85,86]. On the other hand, cap-dependent translation of vesicular stomatitis virus mRNAs requires eL40/Rpl40 [87].

Several studies reveal that changes in ribosomal protein composition are critical for proper biological development. First, deficiency or mutational changes in several ribosomal proteins cause several congenital diseases, including Diamond–Blackfan Anemia and Schwachman–Diamond Syndrome [88,89]. Second, the reduced translation of Hox mRNA in eL38/Rpl38 loss-of-function mutations results in improper axial skeletal patterning [84,90]. Third, the abundance of uL1/Rpl10A, eS25/Rps25, and P1/P2 proteins in polysomes decreases during stages of mesoderm development, presumably due to preferential recruitment of subunits lacking these proteins. Fourth, eS25/Rps25 is implicated in the development of neurogenerative diseases. These diseases are associated with the buildup of amyloid proteins translated by unconventional non-AUG initiation from expanded intronic repeat sequences in the C9orf72 and other genes. Translation of such pathogenic repeats in a yeast model system indicates that eS25/Rps25 is required for their synthesis [91]. Fifth, the abundance of most r-proteins and many ribosome-associated proteins belong to gene ontologies for translation and protein folding change during quiescence and stress [92,93]. This suggests that the ribosomal protein composition helps ensure proper developmental regulation of protein synthesis, a notion supported by the fact that an uL1/Rpl10A loss-of-function mutant decreases the translation of mesoderm regulators, including proteins from the Wnt pathway [94]. This conclusion is also supported by cryo-electron microscopy demonstrating that shifting the carbon source of a yeast culture from glucose to glycerol carbon is accompanied by an increase in the number of ribosomes that lack several r-proteins at the entrance and exit for tRNA, including uL16/Rpl10, eS1/Rps1, uS11/Rps14, and eS26/Rps26 [95].

Mathematical modeling suggests another level of ribosomal regulation of translation. The model shows that reducing total translation capacity due to limited ribosome production or r-protein mutation may alter the ranking of individual mRNAs in the competition for initiation-competent ribosomal subunits. The model has not been tested experimentally, but could contribute to regulating the translatome in parallel with the effects of ribosome r-protein composition [96].

### 4.2. Ribosomal Protein Paralogous Genes

Many r-proteins are encoded by two or more paralogous genes, in some cases called A and B versions, and in others “ribosomal protein” and “ribosomal protein-like” proteins. Some r-protein variants are encoded by autosomal (retro)genes, and others originate from alternate splicing [97,98,99]. Numerous examples in both bacteria and eukaryotes demonstrate that the ribosomal content of the paralogs changes dynamically during stress and development, altering the ribosome preference for binding to specific mRNAs (Figure 2). r-Protein paralog composition often changes during development and altered growth conditions. For example, the expression of the paralogs of r-proteins bL31 and bL36 switch as *E. coli* enters the stationary phase and back during the return to active growth, and the paralogs of bL31 have different effects on reading frame maintenance [100,101]. During *Bacillus subtilis* sporulation, the expressions of three r-proteins switch from zinc-binding (eL31/RpmE, eL33/RpmGA, and uS14/RpsN) to zinc-independent (RpmEB, RpmGC, RpsNB) paralogs, and a triple knock-out of the Zn-independent versions results in a delay in sporulation and a spatial shift in translation activity [102]. Furthermore, mass spectroscopic analysis shows that switching the carbon source in yeast culture from glucose to glycerol triggers a change in the proportion of the eL8/Rpl8A and B paralogs, which are not functionally interchangeable [103]. Similarly, the ribosome paralog composition changes during *Arabidopsis* adaptation to cold conditions, coordinated with apparent changes in the ribosome structure [104].

Enhanced expression of specific paralogs is also essential for sexual development. Mouse male germ development depends on ribosomal incorporation of eL39L/RPL39L, which modifies the shape of the ribosome exit tunnel of the 60S ribosomal subunit, thereby presumably changing the initial folding of nascent proteins [105]. In Drosophila, the paralog composition varies across tissues, and switches between eL22/Rpl22 and eL22L/Rpl22L have been associated with both eye and testes development [106,107]. Furthermore, structural characterization of ribosome populations with heterogenous r-protein paralog composition in ovaries and testes suggests that differences in the ribosome surface may impact ribosomal translation [108,109].

The ribosomal composition of r-protein paralogs is also involved in the fate of tumor cells. For example, a loss of eL22/Rpl22 accelerates the expression of eL22L/Rpl22L1, which is a driver of cell proliferation and anchorage-independent growth in colorectal cancer [110]. The expression of r-protein paralogs may vary with the position of the tumor. In the nucleus of glioblastoma, splicing of the Rpl22L mRNA favors the expression of the 1b form, which interacts extra-ribosomally with the long non-coding RNA ncMALAT and promotes its degradation. In contrast, the 1a form is favored in the tumor periphery, where it is incorporated in ribosomes, stimulating the translation of several proteins, including p53 [111].

Interestingly, the paralogous effect may not be limited to the proteins themselves (see also below). Ribosomes containing murine eS27/Rps27 or eS27L/Rps27L associate with different mRNAs, and deleting either gene causes development lethality [112]. However, animals expressing eS27/Rps27 protein from the endogenous eS27/Rps27L locus or vice versa develop normally, suggesting that features other than the proteins themselves are critical for development. Similarly, uL16/Rpl10 and uL16L/Rpl10L substitute for each other during meiotic sex chromosome inactivation in mice [99]. One might speculate that the DNA sequence contains unidentified signals beyond protein coding.

The phenomenon of paralogous r-protein genes is particularly well-developed in budding and fission yeasts and *Mucoromycota* [113]. It is thought that, in budding yeast, the r-protein paralogs originate from whole genome duplication followed by evolution, resulting in the formation of many pairs of paralogous r-protein [114,115]. Thus, *S. cerevisiae* has paralogous genes for 59 of the 79 r-proteins, while few r-protein genes are duplicated in *Klyuveromyces lactis* and *Candida albicans*, both of which are upstream of the genome duplication event (Table 2). The r-protein gene duplication in *Schizosaccharomyces pombe* may have occurred by retrotransposition [113] (Table 2). Interestingly, the genes encoding the 60S ribosomal “stalk proteins” P1 and P2 are duplicated in all three species.

Most amino acid substitutions in paralogous r-proteins are conservative, as might be expected because both paralogous proteins must be accommodated in the tightly packed ribosomal landscape (Appendix A). Non-conservative amino acid substitutions are almost all in loops close to the ribosome surface and, thus, are not likely to cause major changes in ribosome structure. Furthermore, 21 of the 59 paralogous gene pairs encode identical protein sequences (Table 2 and Appendix A). The high degree of similarity/identity between paralogous r-proteins explains why maintaining both paralogs in a pair is optional for growth. Also, the deletion of one paralog can be compensated for by an increased expression of the other [116,117].

Nevertheless, paralogous gene deletions affect mRNA localization, ribosomal assembly, drug resistance, and mitochondrial formation [93,118,119,120], even when synonymous. It cannot be excluded that such phenotypic changes are due to the selection of secondary mutations after deleting one of the paralogs. On the other hand, the abundance of mRNAs from some pairs of paralogous genes, including two pairs of synonymous paralogs, changes differentially in response to conditional blockage of ribosome formation or function [121]. Since these changes occur over a short period, secondary mutations are unlikely to have been selected, giving weight to the notion that the expression of different paralogs affects the cell physiology in separate ways. For example, many r-proteins are modified post-transcriptionally by acetylation, phosphorylation, or alkylation [122,123,124], and one might speculate that signals for posttranslational modification are laid down in the nucleotide sequences. Furthermore, recent results suggest that the 3′ UTR of r-protein messengers may bind paralog-specific RNA-protein complexes that could contribute to the phenotype of the resulting ribosome [117].

The mechanism of paralog switching is complex. Nucleosome positioning and assembly of the translation complex regulate the transcription activity in yeast, but details for paralogous pairs are not yet elucidated [125,126]. Furthermore, genetic evidence suggests that introns may contribute to the regulation, perhaps through pairing with promoter regions [127,128], but not all paralogous r-protein genes have introns. Recent results indicate that, during spermatogenesis in *Drosophila*, a cognate circular RNA mediates the balance between two paralogs of eL22/RPL22. The complex of eL22/RPL22 and the circular RNA is a repressor of the eL22L/RPL22L gene. The concentration of this repressor complex is, in turn, regulated by competition between eL22/RPL22 and eL22L/RPL22L for binding to the circular RNA [98].

## 5. Conclusions

There is now a substantial and growing body of investigations focused on dissecting the ribosome structure and the functions of individual components. The results clearly connect specific ribosomal features to individual translation steps, refuting the classical view of the ribosome as a single monolithic structure. Modification of rRNA sequence, folding, and posttranscriptional modification, as well as the ribosomal content of ribosomal proteins, all contribute to the ribosome specificity for individual mRNAs and translation initiation sites. Translation accuracy, elongation through specific mRNA sequences, and co-translational peptide folding and modification also depend on variants of ribosomal components.

Importantly, ribosome plasticity is correlated with development phases and the imposition of changing nutritional or stress conditions. This implicates the ribosome as an active component in the panoply of translation regulators that ensure proper cellular functions. In fact, ribosomes may be the final target of some signaling pathways. Despite this significant progress, the mechanisms remain largely opaque. How are the dynamic three-dimensional ribosome changes accomplished, and how do the switches cause adjustments to the ribosome functional state?

Finally, the phenomenon of ribosome functional dynamics is frequently referred to as “specialized ribosomes”. This could be taken to mean that some ribosomes only translate proteins encoded by a single gene. More likely, ribosomal modifications beget adjustment of chemical constants affecting the ribosome’s preference for translating multiple mRNAs and ORFs. Thus, the term “ribosome types” may better describe the phenomenon.

## Figures and Tables

**Figure 1 ijms-25-11186-f001:**
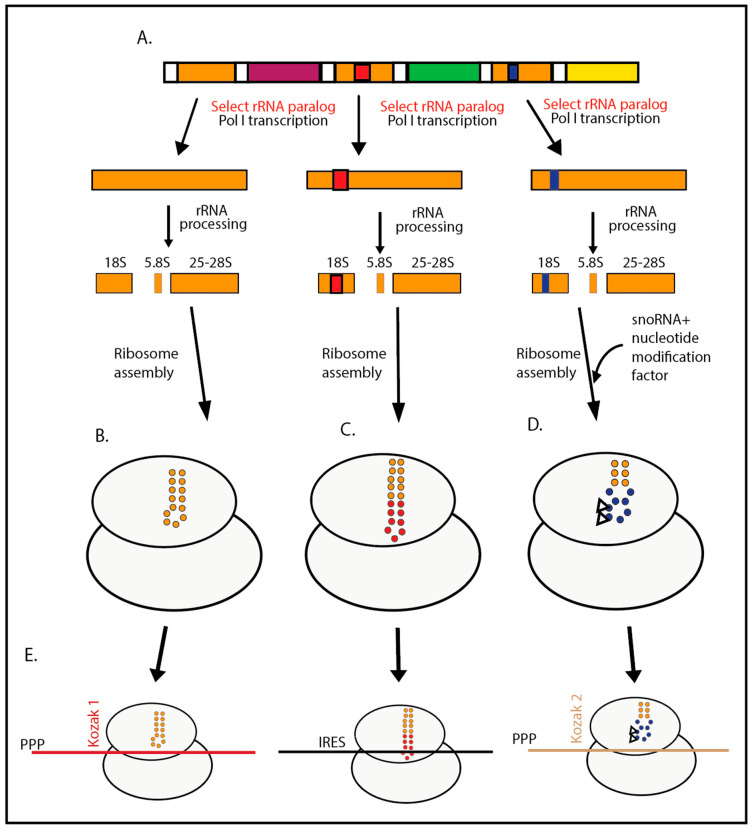
Schematic illustration of the effect of rRNA expansion sequences on the selection of translation initiation sites. (**A**) Array of rRNA genes. The orange, purple, green, and yellow boxes illustrate different versions of rRNA genes. Nucleotide substitutions and indels in the orange gene are illustrated by the blue and red segments in the orange genes in the middle and on the right. Regulatory mechanisms for RNA polymerase I transcription determine which specific rRNA gene(s) are transcribed. (**B**–**D**) Assembled ribosomes, each containing an expansion sequence folded into a hairpin (the individual nucleotides are symbolized by circles). Relative to the orange gene (**B**) (orange nucleotides), (**C**) has an indel elongating the hairpin (red nucleotides), and (**D**) contains nucleotide substitutions that generate a target for snoRNA-mediated posttranscriptional modification (blue nucleotides), shown by the triangles. All other details of the ribosome structure are omitted to promote clarity. (**E**) Translation initiation complexes containing different mRNAs (shown in different colors) selected under the influences of the different hairpins in the (**B**–**D**) ribosomes.

**Figure 2 ijms-25-11186-f002:**
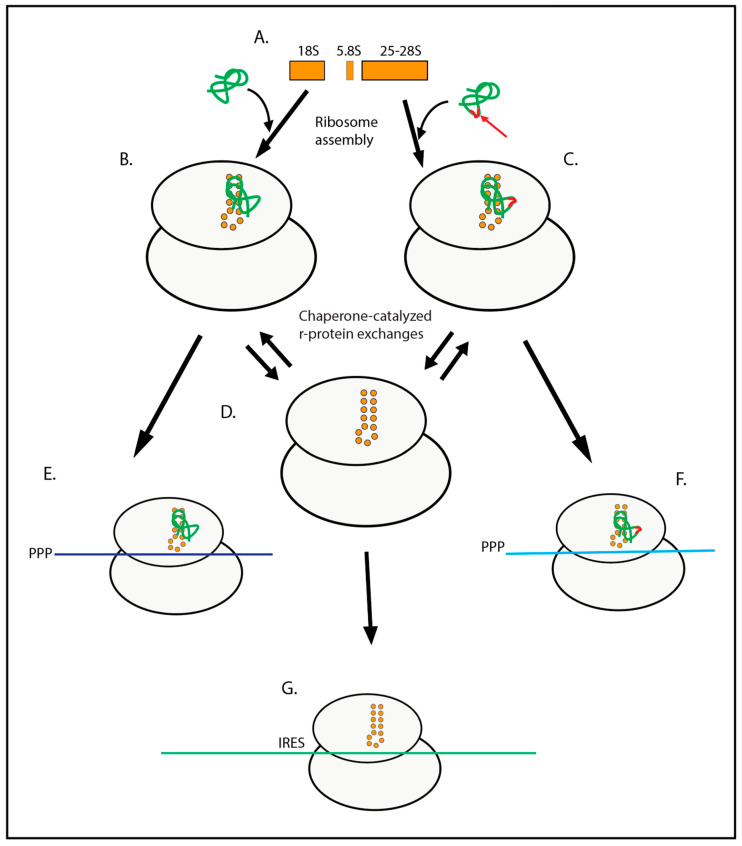
Ribosomes missing specific r-proteins or containing different protein paralogs have varying specificity for binding to particular mRNAs. (**A**) rRNA components (orange). (**B**) Assembled ribosome containing an r-protein (green) binding to an rRNA hairpin symbolized by the orange circles. Other rRNA and r-protein particulars are omitted for the sake of clarity. (**C**) Assembled ribosome containing a paralog of the green r-protein with sequence changes in the red loop (highlighted by the red arrow). The triangular configuration of (**B**–**D**) illustrates the reciprocal replacement of the two r-protein paralogs, in which a ribosome missing the r-protein is an intermediate. All other details of the ribosome structure are omitted for clarity. (**E**–**G**) Translation initiation complexes containing different mRNAs (indicated by purple, cyan, and green colors) selected under the influence of the ribosomal r-protein content.

**Table 2 ijms-25-11186-t002:** Number of paralogous ribosomal protein genes in *Klyuveromyces lactis* (Kl), *Candida albicans* (Ca), *Saccharomyces cerevisiae* (Sc) and *Schizosaccharomyces pombe* (Sp).

Universal Protein Name [13]	Kl gene Name	Paralogous Genes in Kl	Ca Gene Names	Paralogous Genes in Ca	Sc Gene Names	Paralogous Genes in Sc	Sp Gene Names	Paralogous Genes in Sp	Universal Protein Name	Kl Gene Name	Paralogous Genes in Kl	Ca Gene Names	Paralogous Genes in Ca	Sc Gene Names	Paralogous Genes in Sc	Sp Gene Names	Paralogous Genes in Sp
eS1	B05060	1	RPS1	1	RPS1	2	rps3A	2	eL6	B04686	1	RPL6	1	RPL6	2	rpl6	1
uS2	E16171	1	YST1	1	RPS0	2	rpsA-1	2	eL8	E00573	1	RPL8	2	RPL8	2	rpl7A	1
uS3	D08305	1	RPS3	1	RPS3	1	rps3	1	uL11	A01903	1	RPL12	1	RPL12	2	rpl12	2
uS4	E23673	1	RPS9B	1	RPS9	2	rps9	2	uL13	F04675	1	RPL16	1	RPL16	2	rpl13A	3
eS4	B03652	1	RPS4	2	RPS4	2	rps4	3	eL13	E22001	1	RPL13	1	RPL13	2	rpl13	1
uS5	F09812	1	RPS21	1	RPS2	1	rps2	1	uL14	E06997	1	RPL23	1	RPL23	2	rpl23	2
eS6	E24047	1	RPS6	1	RPS6	2	rps6	2	eL14	B13409	1	RPL14	1	RPL14	2	rpl14	1
uS7	D10659	1	RPS5	1	RPS5	1	rps5	2	uL15	F06831	1	RPL28	1	RPL28	1	rpl28/rpl27A	2
eS7	C13519	1	RPS7A	1	RPS7	2	rps7	1	eL15	F17633	1	RPL15	1	RPL15	2	rpl15	2
uS8	B07931/B07601	2	RPS22	1	RPS22	2	rps15A	2	uL16	D05643	1	RPL10	1	RPL10	1	rpl10	2
eS8	E20461	1	RPS8A	1	RPS8	2	rps8	2	uL18	D06941	1	RPL5	1	RPL5	1	rpl5	2
uS9	E21979	1	RPS16A	1	RPS16	2	rps16	2	eL18	A07227	1	RPL18	1	RPL18	2	rpl18	2
uS10	F25542	1	RPS20	1	RPS20	1	rps20	1	eL19	E12453	1	RPL19	1	RPL19	2	rpl19	2
eS10	B08173	1	RPS10	1	RPS10	2	rps10	2	eL20	F08657	1	RPL20	1	RPL20	2	rpl18A	2
uS11	B07623	1	RPS14	1	RPS14	2	rps14	2	eL21	E23651	1	RPL21	1	RPL21	2	rpl21	2
uS12	B11231	1	RPS23	1	RPS23	2	rps23	2	uL22	A06336	1	RPL17	1	RPL17	2	rpl17	2
eS12	F00352	1	RPS12	1	RPS12	1	rps12	2	eL22	D05181	1	RPL22	1	RPL22	2	rpl22	1
uS13	B01562	1	RPS18	1	RPS18	2	rps18	2	eL23	E07481	1	RPL25	1	RPL25	1	rpl23A	2
uS14	E05127	1	RPS29	1	RPS29	2	rps29	1	uL24	B05742	1	RPL26	1	RPL26	2	rpl26	1
uS15	F18040	1	RPS13	1	RPS13	1	rps13	1	eL24	E10869	1	RPL24	1	RPL24	2	rpl24	2
uS17	A10483	1	RPS11	1	RPS11	2	rps11	2	eL27	E03455	1	RPL27A	1	RPL27	2	rpl27	2
eS17	B01474	1	RPS17	1	RPS17	2	rps17	2	uL29	F05247	1	RPL35	1	RPL35	2	rpl35	1
uS19	F07843	1	RPS15	1	RPS15	1	rps15	2	eL29	D16071	1	RPL29	1	RPL29	1	rpl29	1
eS19	A07194	1	RPS19	1	RPS19	2	rps19	2	uL30	D03410	1	RPL7	1	RPL7	2	rpl7	3
eS21	A05700	1	RPS21	1	RPS21	2	rps21	1	eL30	E10847	1	RPL30	1	RPL30	1	rpl30	2
eS24	C07755	1	RPS24	1	RPS24	2	rps24	2	eL31	B02937	1	RPL31	1	RPL31	2	rpl31	1
eS25	B06182	1	RPS25	1	RPS25	2	rps25	2	eL32	E06843	1	RPL32	1	RPL32	1	rpl32	2
eS26	D05115	1	RPS26	1	RPS26	2	rps26	2	eL33	D07405	1	RPL33	1	RPL33	2	rpl35A	2
eS27	E11661	1	RPS27	2	RPS27	2	rps27	1	eL34	C08371	1	RPL34	1	RPL34	2	rpl34	2
eS28	RPS33#	1	RPS28	1	RPS28	2	rps28	2	eL36	D12540	1	RPL36	1	RPL36	2	rpl36	2
eS30	C04809	1	RPS30	1	RPS30	2	rps30	2	eL37	C01870	1	RPL37	1	RPL37	2	rpl37	2
eS31	D18304	1	UBI3	1	RPS31	1	rps31	2	eL38	C18216	1	RPL38	1	RPL38	1	rpl38	2
RACK	E12277	1	ASC1	1	ASC1	1	cpc2	1	eL39			RPL39	1	RPL39	1	rpl39	1
									eL40	A00616/D18304	2	RPL40	1	RPL40	2	rpl40	2
uL1	B02002	1	RPL10A	1	RPL1	2	rpl10A	2	eL41					RPL41	2	rpl41	2
uL2	D16027	1	RPL2	1	RPL2	2	rpl8	3	eL42	D07832	1	RPL42	1	RPL42	2	rpl36A	1
uL3	F16511	1	RPL3	1	RPL3	1	rpl3	2	eL43	E05941	1	RPL43	1	RPL43	2	rpl37A	1
uL4	B07139	1	RPL4	1	RPL4	2	rpl4	2	uL10	B05918	1		1	RPP0	1	rpl0	1
uL5	F08261	1	RPL11	1	RPL11	2	rpl11	2	P1A	E22023/F05555	2	RPP1	2	RPP1	2	rplP1	3
uL6	E09109/F04499	2	RPL9B	1	RPL9	2	rpl9	2	P2A	F07865/E02861	2	RPP2	2	RPP2	2	rplP2	3
Color code for number of amino acid differences between paralogous proteins:										Nill	≤3	4–10	11–15	≥16			
Sources; all URLs were last accessed on 22 December 2023																	
	K lactis (Kl)		https://www.ncbi.nlm.nih.gov/datasets/genome/GCF_000002515.2/														
			https://www.ncbi.nlm.nih.gov/nuccore/NC_003423.3														
			https://www.ncbi.nlm.nih.gov/nuccore/NC_003421.2														
	C albicans (Ca)		http://www.candidagenome.org/														
	S cerevisiae (Sc)		https://www.yeastgenome.org/														
	S pombe (Sp)		https://www.ncbi.nlm.nih.gov/nuccore/NC_003424.3														

## Data Availability

Not applicable.

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
