# Peer review of "Ribosome Structural Changes Dynamically Affect Ribosome Function"

_ijms, 2024, doi:10.3390/ijms252011186_

Round 1
Reviewer 1 Report
Comments and Suggestions for Authors
The manuscript is devoted to the description of variability among ribosomal proteins and genes. The title does not match the content of the manuscript since only some special cases of diversity is mentioned, so it should be more specialized and be more concrete. The text should be structured and expanded as well as illustrative materials.
Here some points that might be edited:
Section "Introduction" should be expanded with some general description of ribosome structure and function, any historical or evolutionary aspects could also be included.
Section "Basic organization of ribosomes": the information about general structure of small and large subunit, rRNA gene (organization of nucleotide sequences, for example), ribosomal proteins (types) should be included.
Table 1: it would be better to turn it into album format.
Figures 1, 2: font size should be enlarged and sharpen. There is no Figure 1 G but it was mentioned in the text (pages 3 and 8).
Probably, Section 4.3 "R-protein paralogs in fungi" could be included in the Section 4.2 "R-protein paralogous genes".
In "Conclusions" presented examples of diversity should be summarized and, probably, related to flexibility and diversity in ribosome functioning or, at least, author should suppose his own point of view on the structural diversity and its consequences.
There is no Supplementary Table.
Author Response
Please see attched

Reviewer 2 Report
Comments and Suggestions for Authors
The manuscript comes across very rushed, with little care taken in its authorship. For example, the Abstract is just over 2 lines of text - this is not acceptable. And similarly, the Introduction is 1 paragraph in length - again, not an acceptable length.
Numerous sentences must have their wording improved in order for the manuscript to be of an acceptable standard / understandable to the reader. Throughout the manuscript little background information is supplied which would be required by most in order to provide context.
Most of my highlighted issues (annotated PDF) are to do with rushed text, or a lack of information provision. This needs to be improved throughout the manuscript. Figure 1 and 2 are of a low quality and therefore require improvement to be at a standard suitable for publication. Also, considering how brief this manuscript is, what is the only Table provided as a Supp Table.
Please use my annotated PDF to address all my concerns regarding your manuscript.

Needs improvement throughout the manuscript
Round 2
Reviewer 1 Report
Comments and Suggestions for Authors
The author edited and improved the manuscript significantly. But there are some details that should be corrected.
Section "3.1 The expansion sequences are important for ribosome assembly and mRNA binding":
"In bacteria, the rDNA variants are organized as single transcription units located at separate genome positions, but in eukaryotes, they are arranged in one or more arrays of tandem repeats."
In the light of the recent publication (Brewer TE, Albertsen M, Edwards A, Kirkegaard RH, Rocha EPC, Fierer N. Unlinked rRNA genes are widespread among bacteria and archaea. ISME J. 2020 Feb;14(2):597-608. doi: 10.1038/s41396-019-0552-3.) it should be mentioned about unlinked rRNA genes in prokaryotes as well as tha in bacteria rRNA genes are organized in one operon and its copy number varies among bacterial genomes (1-15).
Figure 1A: please, provide definition of green (dark and light) and yellow boxes.
Page 7: The sentence "Thus, ribosomes missing uS2/Rps2 or bS21/Rps21 initiate on the proper start at the 5’ end of the bacteriophage lambda cI mRNA." is required additional explanation or some previous description should be included.
Figure 2 is difficult for understanding "Principle of the effect of r-proteins on the selection of translation initiation sites"
Pages 8, 9: some details of r-proteins would be appreciated, not just their names as "uL1/Rpl10A, eL38/Rps38, eS7/Rps7, and eS25/Rps25"/
Page 9: "Many r-proteins are encoded by two or more paralogous genes. Some rprotein variants are encoded by autosomal (retro)genes and others, called “ribosomal protein-like” proteins (RpxYL or RpxY-Like), originate from alternate splicing." Please, provide references.
Table 2 is overwhelmed with information. Probably, it could be moved to Supplemental materials and some concluding sentences could be added to the text.
Table 3 should be re-formatted, probably information about "Difference in aa sequence" can be ommited or moved to Supplemental materials.
Reviewer 2 Report
Comments and Suggestions for Authors
Dear author,
Please address the next round of requested revisions I have made to your manuscript - please refer to the attached annotated MS file.
The manuscript has improved on its first version, but some additional tidying up would be highly beneficial to the overall impact of your manuscript.
Thank you kindly for your efforts with generating the revised manuscript version - they are appreciated.
Regards, Andy

Standard of English language has been improved on the original version - a MINOR REVISION of the text is now required for further improvement.
